# Variable Induction of Pro-Inflammatory Cytokines by Commercial SARS CoV-2 Spike Protein Reagents: Potential Impacts of LPS on In Vitro Modeling and Pathogenic Mechanisms In Vivo

**DOI:** 10.3390/ijms22147540

**Published:** 2021-07-14

**Authors:** Weiming Ouyang, Tao Xie, Hui Fang, Chunling Gao, Tzanko Stantchev, Kathleen A. Clouse, Kun Yuan, Tongzhong Ju, David M. Frucht

**Affiliations:** 1Division of Biotechnology Review and Research II, Office of Biotechnology Products, Office of Pharmaceutical Quality, Center for Drug Evaluation and Research, U.S. Food and Drug Administration, Silver Spring, MD 20993, USA; weiming.ouyang@fda.hhs.gov (W.O.); tao.xie@fda.hhs.gov (T.X.); hui.fang@fda.hhs.gov (H.F.); 2Division of Biotechnology Review and Research I, Office of Biotechnology Products, Office of Pharmaceutical Quality, Center for Drug Evaluation and Research, U.S. Food and Drug Administration, Silver Spring, MD 20993, USA; chun.gao@fda.hhs.gov (C.G.); tzanko.stantchev@fda.hhs.gov (T.S.); kathleen.clouse@fda.hhs.gov (K.A.C.); 3Division of Biotechnology Review and Research III, Office of Biotechnology Products, Office of Pharmaceutical Quality, Center for Drug Evaluation and Research, U.S. Food and Drug Administration, Silver Spring, MD 20993, USA; kun.yuan@fda.hhs.gov (K.Y.); tongzhong.ju@fda.hhs.gov (T.J.)

**Keywords:** SARS CoV-2, COVID-19, spike protein, ACE2, proinflammatory cytokine, endotoxin

## Abstract

Proinflammatory cytokine production following infection with severe acute respiratory syndrome coronavirus 2 (SARS CoV-2) is associated with poor clinical outcomes. Like SARS CoV-1, SARS CoV-2 enters host cells via its spike protein, which attaches to angiotensin-converting enzyme 2 (ACE2). As SARS CoV-1 spike protein is reported to induce cytokine production, we hypothesized that this pathway could be a shared mechanism underlying pathogenic immune responses. We herein compared the capabilities of Middle East Respiratory Syndrome (MERS), SARS CoV-1 and SARS CoV-2 spike proteins to induce cytokine expression in human peripheral blood mononuclear cells (PBMC). We observed that only specific commercial lots of SARS CoV-2 induce cytokine production. Surprisingly, recombinant SARS CoV-2 spike proteins from different vendors and batches exhibited different patterns of cytokine induction, and these activities were not inhibited by blockade of spike protein-ACE2 binding using either soluble ACE2 or neutralizing anti-S1 antibody. Moreover, commercial spike protein reagents contained varying levels of lipopolysaccharide (LPS), which correlated directly with their abilities to induce cytokine production. The LPS inhibitor, polymyxin B, blocked this cytokine induction activity. In addition, SARS CoV-2 spike protein avidly bound soluble LPS in vitro, rendering it a cytokine inducer. These results not only suggest caution in monitoring the purity of SARS CoV-2 spike protein reagents, but they indicate the possibility that interactions of SARS CoV-2 spike protein with LPS from commensal bacteria in virally infected mucosal tissues could promote pathogenic inflammatory cytokine production.

## 1. Introduction

SARS CoV-2 is the etiological agent responsible for a global pandemic that, as of 26 May 2021, had led to 168.8 hundred million cases and over 3.5 million deaths [1]. This virus is a member of the beta-coronavirus family and shares 79.5% sequence homology with SARS CoV-1 [2], which emerged in Southeast Asia in 2003, causing a limited epidemic [3,4,5]. Similar to SARS CoV-1 [6], SARS CoV-2 enters host cells via its surface spike protein, which binds ACE2 on host cells [7,8,9]. SARS CoV-2 spike protein exists as a trimer, with three receptor-binding S1 heads located on top of a trimeric membrane fusion S2 stalk. The receptor-binding domain (RBD) of the S1 subunit specifically recognizes ACE2. Protease-mediated cleavage of the spike protein at the S1/S2 boundary leads to S1 dissociation and S2-mediated membrane fusion [10,11].

During viral infection, host cells recognize the pathogen-associated molecular patterns (PAMPs) of the virus. For example, viral double-stranded RNA (dsRNA) is sensed by the pathogen recognition receptors (PRRs), which include the retinoic acid-inducible gene I (RIG-I)-like receptors (RLRs; RIG-I and melanoma differentiation-associated protein 5 [MDA-5]) [12,13] and the toll-like receptors (TLRs) [14,15,16]. Recognition of SARS CoV-2 viral RNAs by these PRRs triggers the expression of type I IFN and other proinflammatory cytokines [17,18]. IFN production during the early phase of viral infection is critical for the host to restrain replication of the virus. During evolution in their natural hosts (bats), coronaviruses have acquired the ability to suppress the early-phase IFN response by multiple mechanisms, including shielding of the dsRNA in double-membrane vesicles, which prevent the sensing by PRRs, inhibiting host protein synthesis, and disrupting essential signal transduction [17,19,20,21,22,23]. While IFN production is blocked in most cell types following coronavirus infection, notable exceptions are plasmacytoid dendritic cells (pDCs), which sense single-strand RNA via toll-like receptor 7 (TLR-7) and express high levels of type I IFN in response to coronavirus infection [19]. A delayed IFN response, together with a high load of virus and proinflammatory cytokines, causes an unbalanced immune state, promoting the progression of COVID-19 disease [24,25,26,27,28,29,30,31,32]. Severe COVID-19 patients exhibit high levels of pro-inflammatory cytokines, lymphopenia, T cell exhaustion and increased neutrophil-to-lymphocyte ratios, which correlate with poor clinical outcomes.

SARS CoV-1 and SARS CoV-2 infect host cells by a similar mechanism [7,8]. However, the viral shedding pattern and disease progression following infection differ between these two viruses [33]. Also, SARS CoV-2 causes a higher viral load than SARS CoV-1 during the early phase of infection [33], but the reason for this difference is not clear. Interestingly, SARS CoV-1 spike protein has been reported to induce cytokine responses following its direct binding to ACE2 [34]. Given that SARS CoV-2 spike protein has a higher binding affinity to ACE2 than SARS CoV-1 spike protein [9,10], we reasoned that SARS CoV-2 spike protein might induce a more robust cytokine response than SARS CoV-1 in human PBMC. In this study, we report that some commercial lots of SARS CoV-2 spike protein-containing fusion proteins, but not lots of SARS CoV-1 or MERS spike protein-containing fusion proteins, stimulate human PBMC to produce proinflammatory cytokines. Unexpectedly, induction of cytokine production by SARS CoV-2 spike fusion proteins consistently exhibits vendor and batch variability, and cytokine inductions are not blocked by soluble ACE2 or neutralizing anti-spike protein antibody. Instead, we find that commercial spike fusion protein-containing reagents contain variable levels of LPS, which, in turn, correlate with their activities to induce cytokine production. The cytokine-inducing activities of spike fusion protein reagents are blocked by polymyxin B, a lipopolysaccharide (LPS) inhibitor. SARS CoV-2 spike protein efficiently captures soluble LPS, rendering it capable of potent pro-inflammatory cytokine induction. Collectively, these findings indicate that co-purifying LPS, but not spike protein itself, induces proinflammatory cytokine responses in human PBMC. These findings not only highlight the need to monitor LPS levels during in vitro and in vivo studies involving recombinant SARS CoV-2 spike protein, but they suggest a potential role for this interaction in virally infected tissues that harbor commensal bacteria.

## 2. Results

### 2.1. SARS CoV-2 Spike Protein-Containing Fusion Proteins Induce Cytokine Production in Human PBMC

To investigate the capability of SARS CoV-2 spike protein to directly induce a cytokine response, we cultured human PBMC from two healthy donors with a commercial SARS CoV-2 S1-Fc fusion protein (Vendor #1, lot 24056-2002-2). Following a 48 h stimulation, levels of IL-1β, IL-6, IL-8, IL-10, IL-12 and TNFα were assessed by flow cytometry using a human inflammatory cytometric bead array (CBA) kit. Because IL-8 levels in some samples were too high to be accurately measured by the CBA assay, they were determined by ELISA. IL-6, IL-8, IL-10 and TNFα were detected in the supernatants of SARS CoV-2 S1-Fc-treated PBMC from both donors, and this induction occurred in a dose-dependent manner (Figure 1). We next directly compared the cytokine-inducing capabilities of MERS S1-Fc and SARS CoV-1 S1-Fc (Vendor #2), as well as SARS CoV-2 S1-Fc and RBD-Fc proteins from two commercial sources (Vendor #1 and Vendor #2). SARS CoV-2 S1-Fc from Vendor #2 and SARS CoV-2 RBD-Fc from Vendor #1 induced robust cytokine responses following 24 h culture, whereas the other fusion protein reagents or an IgG1 monoclonal antibody control (raxibacumab) did not (Figure 2A and Appendix A). Although cytokines were induced by two of the SARS CoV-2 reagents, there was no clear pattern between the manufacturer and/or fusion construct with respect to cytokine induction activity. It was also surprising that SARS CoV-1 S1-Fc fusion protein did not induce the expression of IL-6, IL-8 and TNFα, which is contradictory to a previous report (34). In addition, to exclude a potential impact of FcR crosslinking by the Fc fusion proteins on the cytokine response, we performed experiments using biotinylated S1 and RBD proteins together with plate-bound streptavidin to stimulate human PBMC. As observed with S1 and RBD fusion protein reagents without biotin (Vendor #2), crosslinked S1-biotin but not RBD-biotin induced a robust cytokine response in human PBMC, suggesting that FcR crosslinking did not play a role (Figure 2B and Appendix A).

### 2.2. Inhibition of SARS CoV-2 Spike Fusion Protein-Induced Cytokine Response by Budesonide and MAPK Inhibitors

Early treatment with inhaled budesonide has been reported to dampen inflammation in COVID-19 patients [35]. Consistent with these findings, co-culture of PBMC with budesonide inhibited S1-Fc-induced production of IL-6, IL-8 and TNFα (Figure 3 and Appendix A). As SARS CoV-2 spike protein has been reported to activate the MAPK and NF-κB signaling pathways [36], and the NFAT signaling pathway is involved in cytokine induction in many cells [37,38], we next investigated whether S1-Fc-induced cytokine responses are blocked by inhibitors targeting these signaling pathways. Treatment with NFAT and NF-κB inhibitors did not decrease S1-Fc-induced expression of proinflammatory cytokines (Appendix A). In contrast, JNK1/2 inhibition reduced the induction of IL-6, IL-8 and TNFα; p38 inhibition reduced the induction of IL-6 and TNFα; and Erk1/2 inhibition reduced TNFα induction (Figure 3 and Appendix A).

### 2.3. SARS CoV-2 Spike Fusion Protein-Induced Cytokine Response Is Independent of Its Binding to ACE2

During the investigation of the capability of SARS CoV-2 spike protein to induce a cytokine response, we noticed that SARS CoV-2 S1-Fc protein reagent from Vendor #2, but not Vendor #1 (lot 24529-2003), and RBD-Fc from Vendor #1, but not Vendor #2, induced cytokine responses. In addition, batch variability in PBMC cytokine induction capacity was observed between S1-Fc and RBD-Fc protein lots purchased from Vendor #1 (Appendix A). We reasoned that these inconsistencies could be due to product quality differences affecting spike protein binding to ACE2. To test this hypothesis, we developed an ELISA to measure the binding affinities of spike fusion protein constructs with ACE2. In this assay, ACE2 was coated on a plate, exposed to spike protein-containing test reagents, followed by incubation with HRP-labelled anti-human IgG Fc antibody and substrate development (Appendix A). Absorbance values at 450 nm correlated linearly with the concentrations of S1-Fc protein (Appendix A). Using this ELISA, we observed that SARS CoV-2 S1-Fc and RBD-Fc from both Vendor #1 and Vendor #2, as well as SARS CoV-1 S1-Fc, showed dose-dependent binding to ACE2, which did not correlate with their abilities to induce cytokine production (Figure 4A). Moreover, blockade of S1-Fc binding to ACE2 by soluble ACE2 and an anti-S1 neutralizing antibody did not block S1-induced production of IL-6, IL-8 or TNFα, despite blocking binding of the fusion proteins to ACE2 (Figure 4B,C and Appendix A). Of note, low levels of IL-8 production were observed in PBMC cultured with soluble ACE2 (Figure 4C) or anti-S1 antibody alone (Appendix A). Taken together, our results suggest that the induction of key pro-inflammatory cytokines in human PBMC by certain commercial S1-Fc-fusion proteins occurs independent of ACE2 binding.

### 2.4. LPS Present in Commercial SARS CoV-2 Spike Fusion Preparations Induces Cytokine Production

Spike fusion proteins purchased for our study from commercial vendors were expressed in mammalian cells and purified by a standard protein purification process. Both Vendor #1 and Vendor #2 report a control limit of <1 EU/µg protein for the levels of endotoxin in their protein reagents. One possible explanation for the variation of S1-Fc and RBD-Fc abilities to induce cytokine response was due to variable levels of endotoxin present in these protein preparations. To test this hypothesis, we measured the levels of LPS in the fusion proteins used in our study using a limulus amebocyte lysate (LAL) chromogenic endotoxin quantitation assay. Higher levels of endotoxin were observed in SARS CoV-2 S1-Fc (lot 24056-2002-2) from Vendor #1 and SARS CoV-2 S1-Fc from Vendor #2 (Figure 5A), correlating directly with their strong activities to induce cytokine production (Figure 2). In contrast, the commercial SARS CoV-1 S1-Fc fusion protein had a low level of LPS, in line with its inability to induce a cytokine response (Figure 2). Interestingly, LPS in these SARS CoV-2 S1-Fc protein preparations was not efficiently reduced following treatment of endotoxin removal using Pierce™ high-capacity endotoxin removal spin columns (Figure 5B). The S1-Fc fusion protein still induced a robust cytokine response following two rounds of endotoxin removal treatment (Appendix A). This is consistent with a previous report that LPS binds SARS CoV-2 spike protein with a high affinity [39].

LPS is known to induce a robust cytokine response following a short time period of stimulation [40]. A time-course study showed that a strong cytokine response was induced as early as 3 h following S1-Fc protein stimulation (Figure 5C), consistent with the hypothesis that the induced cytokine responses were due to the presence of LPS. We next obtained additional spike proteins from Vendor #3, which has a 10-fold tighter control for the levels of endotoxin (<0.1 EU/µg protein). The low endotoxin levels in these proteins were confirmed using the LAL chromogenic endotoxin quantitation kit (Appendix A). Consistent with a role for LPS, these protein preparations did not induce a robust cytokine response in human PBMC following 3 h and 24 h stimulations (Appendix A). To further confirm that co-purifying LPS allows spike proteins to induce cytokine production, we cultured human PBMC with Vendor #2 S1-Fc for 3 h together with or without polymyxin B, an LPS inhibitor. S1-Fc-induced production of IL-6, IL-8 and TNFα was inhibited in the presence of polymyxin B (Figure 5D,E and Appendix A). Taken together, these results indicate that co-purifying LPS, but not the spike protein itself, is responsible for the proinflammatory cytokine production in human PBMC.

SARS CoV-2 spike protein efficiently captures LPS, enabling cytokine induction activity.

We next investigated in our study whether SARS CoV-2 S1-Fc and LPS synergistically induce cytokine responses. SARS CoV-2 S1-Fc lot 24529-2003 with a low level of co-purifying endotoxin (Figure 5A) from Vendor #1 was used to stimulate human PBMC together with five serially diluted concentrations of LPS (from 100 mEU/mL–6.25 mEU/mL). The presence of S1-Fc did not change the cytokine expression in human PBMC co-cultured with LPS (Appendix A), suggesting that SARS CoV-2 S1-Fc does not have a synergistic effect on LPS-induced cytokine production in our experimental system. These results differ from the original report describing the LPS/SARS CoV-2 spike protein interaction [37].

Given that LPS is potentially present at sites of SARS CoV-2 infection, and that LPS binds the viral spike protein with a high affinity [39], we reasoned that SARS CoV-2 spike protein may serve as a scaffold to capture soluble environmental LPS, thereby facilitating the access of LPS to innate immune cells in infected tissues. We tested this hypothesis using plate-bound streptavidin, following by incubation with S1-biotin together with serially diluted LPS. Following thorough washing to remove unbound LPS, human PBMC were added to the wells and cultured for 3 h. As shown in Figure 6 and Appendix A, PBMC cultured in the wells with S1-biotin and LPS produced significantly higher levels of IL-6, IL-8 and TNFα than those cultured in wells with LPS alone, supporting our hypothesis that SARS CoV-2 spike protein efficiently captures LPS, forming complexes that activate immune cells to produce proinflammatory cytokines.

## 3. Discussion

Infection with SARS CoV-2 causes no symptoms or mild illness (COVID-19) that resolves spontaneously in most individuals [41,42]. However, a small proportion of patients with COVID-19 progress to a severe disease [41,42]. Severe disease is associated with an imbalanced immune response marked by delayed type I interferon responses to SARS CoV-2 and late-stage overproduction of proinflammatory cytokines [24,25,26,27,28,29,30,31,32]. SARS CoV-1 spike protein is reported to activate immune cells by direct binding to ACE to produce proinflammatory cytokines [34]. As SARS CoV-2 spike protein has a higher binding affinity to ACE2 than SARS CoV-1 spike protein [9], this led to our original hypothesis that SARS CoV-2 spike protein might directly induce a robust cytokine response via its interaction with ACE2. Investigation of this potential mechanism was warranted, because SARS CoV-2 spike protein is the target antigen for SARS CoV-2 vaccines and for neutralizing antibody therapies [7,8]. Moreover, investigation of cytokine induction by engagement of SARS CoV-2 spike protein with ACE2 had the potential to be the basis for a cell-based assay to evaluate the potency of FDA-regulated antibodies targeting SARS CoV-2 spike protein for COVID-19 therapies.

Initial experiments using an S1-Fc fusion protein manufactured by a commercial vendor using transient transfection of Expi293F cells demonstrated that this fusion protein exhibited a dose-dependent induction of cytokine expression in human PBMC, in line with our hypothesis. However, subsequent experiments involving a variety of commercial S1 protein constructs demonstrated inconsistencies between commercial vendors, as well as lot-to-lot variability in cytokine induction activity. Moreover, several orthogonal methods showed that cytokine induction was independent of ACE2 binding. These contradictory findings called into question our original hypothesis.

As the commercial spike fusion proteins used in this study were generated in mammalian cell culture systems, we did not expect significant levels of LPS to be present in these preparations. We nevertheless explored the possibility that variable levels of LPS underlay the inconsistency of these protein preparations to induce cytokine production. Surprisingly, results from LAL assays revealed that the capability of the commercial S1 fusion proteins to induce cytokines directly correlated with their levels of LPS, and the pattern and time course of cytokine induction by these reagents closely matched that of LPS [40]. More convincing evidence supporting the role of the LPS pathway was the observation that S1-Fc-induced cytokine production was blocked by the LPS inhibitor, polymyxin B. In addition, blockade of MAPK signaling pathways, but not the NF-κB pathway, inhibited cytokine induction by S1-Fc. These findings are consistent with a previous report that p38 and JNK1/2 pathways are central to LPS-induced cytokine responses [43]. Interestingly, SARS CoV-1 spike protein-induced IL-6 and TNFα in murine RAW264.7 cells were reported to be mediated by the NF-κB pathway by the She group [34], suggesting that a different mechanism might be involved.

The topic of SARS CoV-2 spike protein-induced cytokine production has been an area of active research, leading to somewhat contradictory conclusions. For example, Hsu et al. reported that SARS CoV-2 spike protein promotes hyper-inflammatory response in human bronchial epithelial cells via ACE2-triggered MAP kinases–NF-κB signaling pathways [36], a result that contrasts with our findings. Although we cannot exclude a role for cell-specific effects, their source for S1-His and RBD-His was Vendor #2, whose controls on endotoxin levels (1 EU/µg protein) allow cytokine-inducing levels of LPS to be present in experimental conditions. Instead, our conclusions regarding a pro-inflammatory role for co-purifying LPS in commercial SARS CoV-2 spike protein preparations are consistent with Petruk et al., who reported that LPS binds to both SARS CoV-1 and SARS CoV-2 spike proteins with high affinity, rendering complexes capable of inducing cytokines, independent of the NF-κB pathway [36]. However, we did not observe the synergistic effect between S1-Fc and LPS reported by Petruk et al., perhaps due to the different spike protein preparations and spike protein:LPS ratios employed in our experiments. Our results also are consistent with the report from the Shirato group, who reported that the SARS CoV-2 spike protein S1 subunit induces pro-inflammatory responses via toll-like receptor 4 signaling in murine and human macrophages, the signaling pathway downstream of LPS recognition [44]. We note that the SARS CoV-1 and SARS CoV-2 spike proteins used in the studies of the She group and the Shirato group, respectively, were prepared using *E. coli* expression systems, where high-affinity binding to LPS would be very likely [34,44].

IL-8 is a neutrophil chemotactic factor. High levels of serum IL-8 and an increased neutrophil-to-lymphocyte ratio are associated with the illness severity of COVID-19 [45]. In this study, we observed that low levels of IL-8 are induced by MERS S1-Fc, SARS CoV-1 S1-Fc, SARS CoV-2 S1-Fc (Vendor #1, lot 24529-2003) and SARS CoV-2 RBD-Fc (Vendor #2 and Vendor #1, lot 25130-2004) (Figure 2 and Appendix A), which do not induce expression of other inflammatory cytokines. In addition, polymyxin B completely blocks S1-Fc-induced IL-6 and TNFα, but not IL-8. These observations may suggest a potential role for S protein–ACE2 interactions in driving IL-8 induction. However, PBMC cultured with soluble ACE2 (Figure 4C) or anti-S1 antibodies alone (Appendix A) also produce low levels of IL-8. Moreover, PBMC cultured with 0.35 mEU/mL of LPS for 24 h or 1.76 mEU/mL of LPS for 3 h results in low-level production of IL-8 (Appendix A). The existence of low levels of LPS in the reagents used to stimulated PBMC in this study is very likely. Taken together, the contribution of the interaction between SARS CoV-2 spike protein and ACE2 to IL-8 induction is likely negligible, although it cannot be fully excluded.

SARS CoV-2 spike protein is highly glycosylated [46]. Glycans of the spike protein can bind C-type lectin receptors (CLRs) on immune cells [47], thereby promoting expression of proinflammatory cytokines. In future studies, we plan to investigate whether glycan profiles differ between various commercial SARS CoV-2 spike protein-containing fusion proteins and the extent to which glycans contribute to cytokine responses induced by these proteins. Our finding that polymyxin B blocks cytokine induction suggests that a direct effect mediated through lectin receptors is unlikely, but the possibility exists that glycosylation of spike protein modulates its binding to LPS.

In summary, our findings confirm that SARS CoV-2 spike protein binds LPS [39], introducing a potentially confounding factor in experimental studies. Unexpectedly, this high-avidity binding has an adverse impact on conclusions generated from experiments involving some commercial SARS-CoV-2 spike protein reagents, even those manufactured in mammalian cell culture systems. Careful monitoring of SARS CoV-2 spike protein reagents for LPS contamination is needed when these reagents are used for in vitro and in vivo studies. The strong binding activity of the spike protein with LPS enables the spike protein to function as a scaffold to capture LPS in vitro and, presumably, in vivo in virally infected mucosal tissues with abundant commensal bacteria. Moreover, as patients with certain COVID-19 risk factors such as diabetes and obesity are characterized by increased levels of Gram-negative commensal bacteria in potential sites of infection such as the lung [48], this interaction could underlie a potential pathogenic mechanism intensifying damaging inflammation in susceptible patient groups.

## 4. Materials and Methods

### 4.1. Human Peripheral Blood Mononuclear Cells

Leukapheresis blood packs from healthy donors were obtained under an institutional review board-approved protocol at the National Institutes of Health. All donors provided written informed consent, and anonymized samples were provided to our laboratory. Cell preparations were performed following the approved laboratory guidelines. Peripheral blood mononuclear cells (PBMC) were prepared by Ficoll density gradient centrifugation. Cells were rested in RPMI 1640 medium supplemented with 10% fetal bovine serum, 2 mM L-glutamine, 100 IU/mL penicillin, 100 µg/mL streptomycin, 1 mM sodium pyruvate, non-essential amino acids, 10 mM HEPES and 55 µM 2-mercaptoethanol overnight before stimulation. Rested PBMC were stimulated with spike Fc fusion protein preparations, with or without plate-bound streptavidin together with biotinylated spike proteins, or LPS captured by plate-bound streptavidin together with biotinylated SARS CoV-2 spike protein S1 subunit.

### 4.2. Reagents

S1-Fc (Catalog # 97-086, lots 24056-2002-2 and 24529-2003), RBD-Fc (Catalog # 10-206, lots 24530-2003 and 25130-2004), S1-biotin (Catalog # 10-208 lot 24598-2003) were obtained from ProSci Inc. (Vendor #1; Poway, CA, USA). MERS and SARS CoV-1 S1-Fc fusion proteins were prepared by Sino Biological (Vendor #2; Beijing, China) under request. S1-Fc (Catalog # 40591-V02H, lot LC14AP1605), RBD-Fc (Catalog # 40592-V02H, lot LC14MC2602), S1-biotin (Catalog # 40591-V27HB, lot LC14AU101), RBD-biotin (Catalog # 40592-V27B-B, lot MF14AP2302), ACE2 (Catalog # 10108-H08H, lot MB14JN0906), anti-S1 neutralizing antibody (Catalog # 40591-MM43, lot HB14AP2001) and anti-S1 non-neutralizing antibody (Catalog # 40591-MM42, lot HB14AP0701) were also purchased from Sino Biological (Vendor #2). S1-Fc (Catalog # 10623-CV, lot DONK0120091) and S1-biotin (Catalog # BT10569, lot DOJW120092) were purchased from R&D Systems (Vendor #3; Minneapolis, MN, USA). Streptavidin (Catalog # 85878-10MG, lot 049K8616V) and LPS from *E. coli* O111:B4 (Catalog # L2630-10MG, lot 114M4009V) were purchased from Sigma (St. Louis, MO, USA). Budesonide (Catalog # S1286), MEK1/2 inhibitor U0126 (Catalog # S1102), JNK1/2 inhibitor SP600125 (Catalog # 1460), p38 inhibitor SB203580 (Catalog # S1076), NFAT inhibitor Cyclosporin A (Catalog # S2286) and NF-κB inhibitor JSH-23 (Catalog # S7351) were purchased from Selleckchem (Houston, TX, USA). NF-κB SN50 cell permeable inhibitory peptide (Catalog # sc-3060) was obtained from Santa Cruz (Dallas, TX, USA). Polymyxin B (Catalog # ICN10056505) was obtained from Fisher Scientific (Waltham, MA, USA). Budesonide was used at a concentration of 50 nM. U0126, SP600125, SB203580 and NF-κB SN50 cell permeable inhibitory peptide were used at final concentrations of 10 µM. Cyclosporin A, JSH-23 and polymyxin B were used at concentrations of 100 nM, 50 µM and 25 µg/mL, respectively.

### 4.3. Cytometric Bead Array of Inflammatory Cytokines

The human inflammatory cytometric bead array (CBA) kit was purchased from BD Biosciences (San Jose, CA, USA). This kit was used to measure the levels of IL-1β, IL-6, IL-8, IL-10, IL-12p70 and TNFα, following the manufacturer’s instructions in supernatants collected from stimulated PBMC. Briefly, 50 µL of mixed beads were incubated with 50 µL of samples or standards and PE-labeled detecting antibodies at room temperature for 3 h. Following one wash of the beads with wash buffer, the beads were resuspended in 100 µL of buffer and analyzed by flow cytometry. Flow cytometry data were analyzed using FlowJo (Tree Star, V10.5.3). The concentrations of cytokines were calculated based on interpolation from a curve generated using the data from the kit’s standards. Cytokine induction was calculated using the following formula: cytokine concentration of stimulated PBMC/cytokine concentration of untreated PBMC.

### 4.4. IL-8 ELISA

The levels of IL-8 in the supernatants were measured using the IL-8 ELISA kit (Catalog # S8000C) purchased from R&D Systems, following the manufacturer’s instructions. Supernatants were diluted based on the fluorescence intensities of IL-8 determined by the CBA assay. Then, 50 μL of diluted samples and standards were added to each well of the precoated ELISA plate and incubated for 2 h at room temperature. Following 4 washes with wash buffer, 100 μL of human IL-8 conjugate was added to each well and incubated for 1 h at room temperature. Following 4 washes with wash buffer, 200 μL of substrate solution were then added to each well and incubated for 30 min at room temperature. Substrate development was stopped by adding 50 μL of stop solution to each well. Absorbance at 450 nm was measured using an ELISA plate reader. The concentrations of IL-8 in supernatants were calculated based on the standard curve and multiplied the dilution factors. IL-8 induction was calculated using the following formula: IL-8 concentration of stimulated PBMC/IL-8 concentration of untreated PBMC.

### 4.5. Spike Protein Binding Assay

An ELISA was developed to measure the binding activities of spike proteins with ACE2. Next, 100 µL of 2 µg/mL ACE2 in PBS were added to each well of a 96-well ELISA plate and incubated at 4 °C overnight. Following 3 washes with PBS, the wells were blocked with 2% BSA in PBS containing 0.5% tween-20 (PBST) at room temperature for 1 h. The wells were then incubated with 100 µL of serially diluted spike protein samples in PBST containing 0.1% BSA at room temperature for 1 h. Following 3 washes with PBST, 100 µL of HRP-labelled anti-human IgG Fc antibody (Catalog # 31413, Thermo Fisher) were added to each well and incubated for 1 h. Then, 100 µL of substrate (Catalog # 34028, Thermo Fisher) were added to each well following 4 washes with PBST and incubated at 37 °C for 20 min. The development was stopped by adding 50 µL of 2 M H_2_SO_4_, and the absorbance at 450 nm was measured using an ELISA plate reader. The relative binding activities were presented as the values of O.D. 450. To assess the blockade of spike protein binding to ACE2 by soluble ACE2 and anti-S1 neutralizing antibodies, 400 ng/mL of S1-Fc protein were preincubated with 2 µg/mL of ACE2 or 1 µg/mL of anti-S1 antibodies at room temperature for 20 min before incubation with plate-bound ACE2.

### 4.6. Endotoxin Removal

Endotoxin removal was performed using 0.25 mL Pierce high-capacity endotoxin removal spin columns (Catalog # 88273, Thermo Fisher), following the manufacturer’s instructions. Briefly, endotoxin removal resin was regenerated by incubation with 0.2 N NaOH in 95% ethanol at room temperature for 2 h, followed by washes with 2 M NaCl, endotoxin-free ultrapure water and endotoxin-free PBS. The regenerated resin was then incubated with S1-Fc spike protein reagents and streptavidin at 4 °C overnight. The protein samples were recovered by centrifugation at 500× *g* for 1 min.

### 4.7. LAL Endotoxin Measurement

The levels of endotoxin in spike protein reagents and streptavidin treated with or without high-capacity endotoxin removal spin columns were measured using the Pierce™ LAL Chromogenic Endotoxin Quantitation Kit (Catalog # 88282, Thermo Fisher, Waltham, MA, USA), following the manufacturer’s instruction. Briefly, 50 µL of diluted protein sample and LPS standards were added to each well of a 96-well plate. The plate was warmed to 37 °C in a water bath, and 50 µL of limulus amebocyte lysate were added to each well and incubated for 10 min following briefly mixing. Then, 100 µL of chromogenic substrate solution (prewarmed to 37 °C) were added to each well, and 100 µL of 25% acetic acid were added to wells to stop the reaction following 6-min substrate development. The absorbance at 405 nm was measured using an ELISA plate reader. The concentrations of endotoxin were calculated based on the standard curve created using the absorbance values of the standards.

### 4.8. Statistical Analyses

All statistical analyses were conducted using Excel software and performed using the two-tailed Student’s *t*-test. The significance threshold was set at *p* < 0.05.

## Figures and Tables

**Figure 1 ijms-22-07540-f001:**
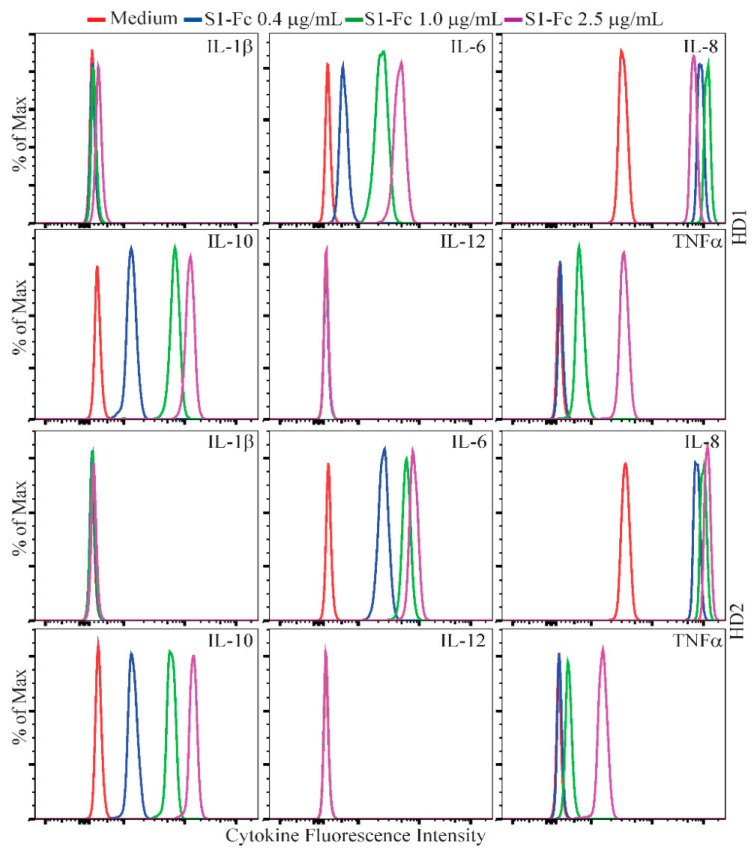
Dose-dependent cytokine responses of human PBMC induced by SARS CoV-2 S1-Fc fusion protein. Rested PBMC from two healthy donors (HD1 and HD2) were cultured for 48 h with 0, 0.4, 1.0 and 2.5 µg/mL of S1-Fc purchased from Vendor #1 (lot 24056-2002-2). The levels of IL-1β, IL-6, IL-8, IL-10, IL-12 and TNFα in the supernatants of cultured PBMC were assessed using the CBA human inflammatory cytokine kit and flow cytometric analysis. Data shown are representative of the flow cytometric results from at least two separate experiments performed on samples from healthy donors, HD1 and HD2.

**Figure 2 ijms-22-07540-f002:**
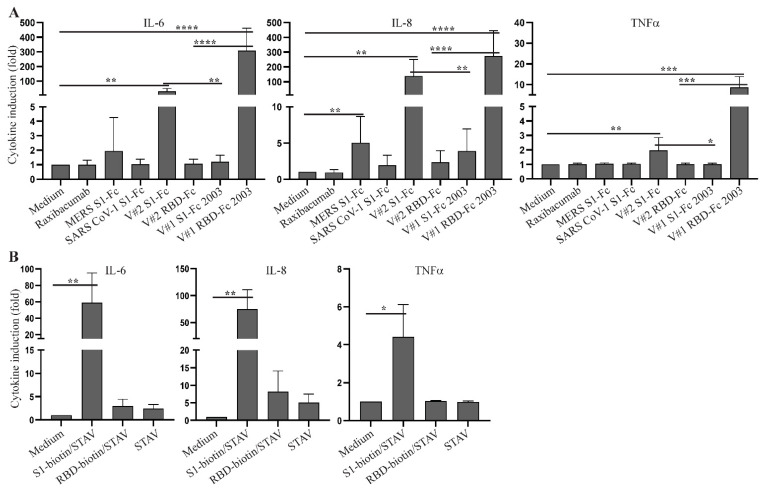
Cytokine responses in human PBMC induced by various commercial coronavirus spike proteins. (**A**) Rested PBMC were cultured with or without 2.0 µg/mL of raxibacumab (human anti-anthrax PA IgG used as a negative control), MERS S1-Fc, SARS CoV-1 S1-Fc, SARS CoV-2 S1-Fc from Vendor #2 (V#2 S1-Fc) and Vendor #1 (lot 24529-2003, V#1 S1-Fc 2003), and RBD-Fc from Vendor #2 (V#2 RBD-Fc) and Vendor #1 (lot 24530-2003, V#1 RBD-Fc 2003) for 24 h. (**B**) Rested PBMC were cultured with or without plate-bound streptavidin (STAV) together with or without S1-biotin and RBD-biotin purchased from Vendor #2. The levels of IL-6 and TNFα were measured using the CBA human inflammatory cytokine kit and flow cytometric analysis. IL-8 levels were measured using an ELISA kit. The concentrations of the cytokines were calculated based on the standard curves, and the induction of cytokines were presented as cytokine induction (fold) calculated using cytokine concentrations and the formula: cytokine concentration of treated PBMC/cytokine concentration of untreated PBMC. Data shown are statistical results (mean ± SE) generated from 8 (**A**) or 3 (**B**) healthy donors. Statistical analyses were performed by Excel using a two-tailed, Student’s *t*-test. *, **, *** and **** depict *p* < 0.05, 0.01, 0.005 and 0.001, respectively.

**Figure 3 ijms-22-07540-f003:**
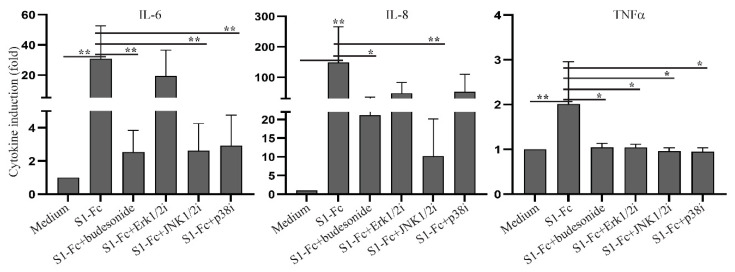
Inhibition of S1-Fc-induced cytokine responses by budesonide and MAPK inhibitors. Rested PBMC were cultured with or without 2.0 µg/mL of S1-Fc from Vendor #2 in the presence or absence of budesonide, U0126 (MEK1/2 inhibitor, Erk1/2i), SP600125 (JNK1/2 inhibitor, JNK1/2i) or SB203580 (p38 inhibitor, p38i) for 24 h. The levels of IL-6, TNFα and IL-8 were measured as described in Figure 2. Data shown are mean cytokine induction indices (fold) ± SE derived from 8 healthy donors, which were calculated using cytokine concentrations measured by the CBA assay (IL-6 and TNFα) or the ELISA (IL-8) and the formula: cytokine concentration of treated PBMC/cytokine concentration of untreated PBMC. Statistical analyses were performed using a two-tailed, Student’s *t*-test. * and ** depict *p* < 0.05 and 0.01, respectively.

**Figure 4 ijms-22-07540-f004:**
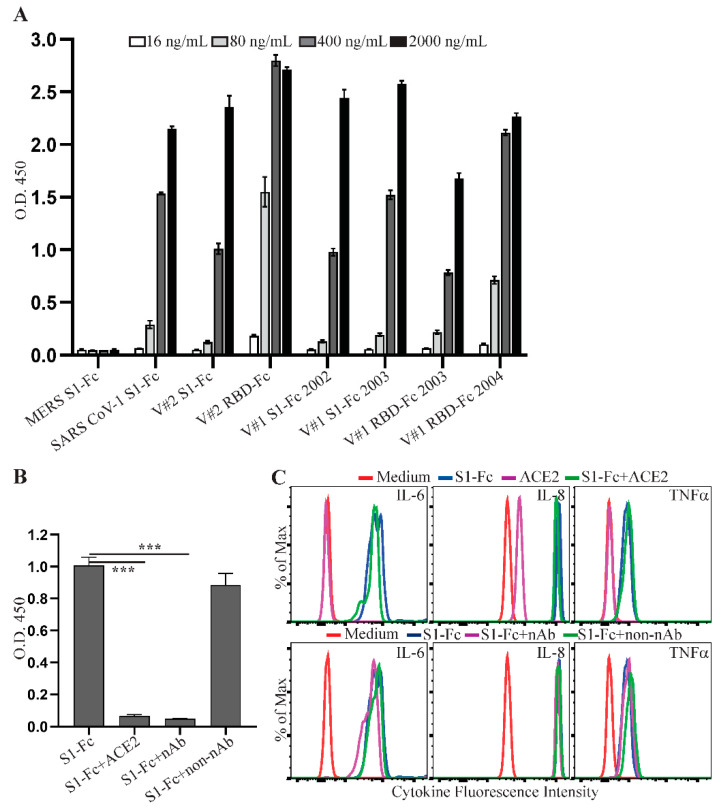
SARS CoV-2 spike protein-induced cytokine production is independent of its binding to ACE2. (**A**) The binding activities of coronaviral spike Fc fusion proteins to ACE2 were measured using an ELISA. The binding activities of 16, 80, 400 and 2000 ng/mL of the indicated proteins were presented as absorbance at 450 nm (O.D. 450). (**B**) Inhibition of S1-Fc binding to ACE2 by soluble ACE2 or a neutralizing anti-S1 antibody; 400 ng/mL of S1-Fc were preincubated with or without 2 µg/mL of soluble ACE2, 1 µg/mL of neutralizing (nAb) or non-neutralizing (non-nAb) anti-S1 antibody before incubation with plate-bound ACE2. Statistical analyses were performed using a two-tailed, Student’s *t*-test. *** depicts *p* < 0.001. (**C**) Rested PBMC were cultured with or without 2.0 µg/mL of S1-Fc in the presence or absence of 5 µg/mL of soluble ACE2, 2 µg/mL of nAb or non-nAb for 24 h. The levels of IL-6, IL-8 and TNFα in the supernatants of cultured PBMC were assessed using the CBA human inflammatory cytokine kit and flow cytometric analysis.

**Figure 5 ijms-22-07540-f005:**
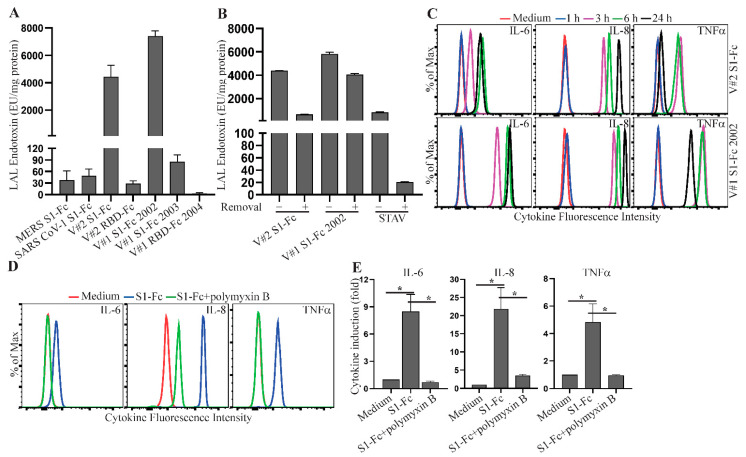
LPS co-purifying with spike protein reagents induces proinflammatory cytokine production. (**A**) MERS S1-Fc, SARS CoV-1 S1-Fc, S1-Fc from Vendor #2 (V#2 S1-Fc), S1-Fc from Vendor #1 (lot 24056-2002-2 (V#1 S1-Fc 2002) and lot 24529-2003 (V#1 S1-Fc 2003)), RBD-Fc from Vendor #2 (V#2 RBD-Fc) and RBD-Fc from Vendor #1 (lot 25130-2004, V#1 RBD-Fc 2004), and (**B**) V#2 S1-Fc, V#1 S1-Fc 2002 and streptavidin (STAV) before and after treatment of endotoxin removal were assessed for the levels of endotoxin using the LAL Chromogenic Endotoxin Quantitation Kit. The concentrations of endotoxin are presented as EU/mg protein. Data shown are mean ± SE of the results from three independent experiments. (**C**) Time course of S1-Fc-induced cytokine responses. Rested PBMC were cultured with or without 2.0 µg/mL of S1-Fc (V#2 S1-Fc and V#1 S1-Fc 2002) for 1, 3, 6 or 24 h. The levels of IL-6, IL-8 and TNFα in the supernatants of cultured PBMC were assessed using the CBA human inflammatory cytokine kit and flow cytometric analysis. (**D**,**E**) Blockade of S1-Fc-induced cytokine response by an LPS inhibitor, polymyxin B. Rested PBMC were cultured with or without 2.0 µg/mL of S1-Fc (V#2 S1-Fc) in the presence or absence of polymyxin B for 3 h. The levels of IL-6, IL-8 and TNFα in the supernatants of cultured PBMC were measured using the CBA human inflammatory cytokine kit and flow cytometric analysis. Data shown in (**D**) are representative of the flow cytometric results from three healthy donors and data shown in (**E**) are mean cytokine induction induces (fold) ± SE, which were calculated using cytokine concentrations and the formula: cytokine concentration of treated PBMC/cytokine concentration of untreated PBMC. Statistical analyses were performed using a two-tailed, Student’s *t*-test. * depicts *p* < 0.05.

**Figure 6 ijms-22-07540-f006:**
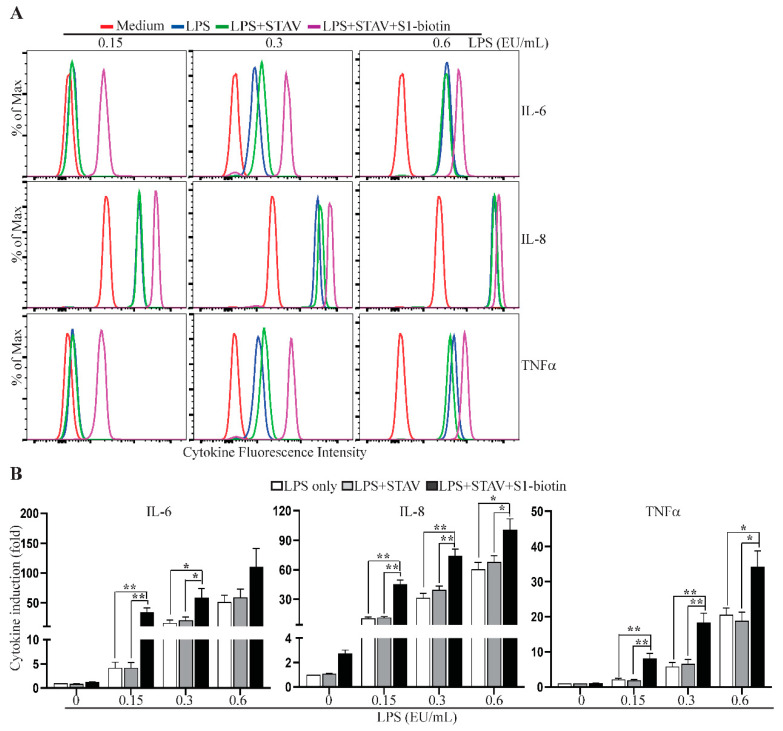
Spike protein captures endotoxin to induce cytokine expression. Rested PBMC were cultured for 3 h in wells, which were coated with or without 100 µL of streptavidin (STAV, 2.0 µg/mL) and S1-biotion from Vendor #1 together with 0.15, 0.3 or 0.6 EU/mL LPS. The levels of IL-6, IL-8 and TNFα in the supernatants of cultured PBMC were measured using the CBA human inflammatory cytokine kit and flow cytometric analysis. The concentrations of IL-8 in the supernatants were also measured using an ELISA kit. Data shown in (**A**) are representative of the flow cytometric results from four healthy donors, and data shown in (**B**) are mean of cytokine induction indices (fold) ± SE derived from four healthy donors, which were calculated using cytokine concentrations measured by the CBA assay (IL-6 and TNFα) or the ELISA (IL-8) and the formula: cytokine concentration of treated PBMC/cytokine concentration of untreated PBMC.. Statistical analyses were performed using two-tailed, Student’s *t*-test. * and ** depict *p* < 0.05 and 0.01, respectively.

## Data Availability

The data presented in this study are available in this article as main figures or Appendix A.

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
