# Peer review of "Variable Induction of Pro-Inflammatory Cytokines by Commercial SARS CoV-2 Spike Protein Reagents: Potential Impacts of LPS on In Vitro Modeling and Pathogenic Mechanisms In Vivo"

_ijms, 2021, doi:10.3390/ijms22147540_

Round 1
Reviewer 1 Report
General comments:
In this paper, Ouyang et al report that careful monitoring of SARS CoV-2 spike protein reagents for LPS contamination is needed as SARS CoV-2 spike protein avidly binds LPS, and propose that the strong binding activity of the spike protein with LPS enables the spike protein to function as a scaffold to capture LPS in vitro and, presumably, in vivo in virally infected mucosal tissues with abundant commensal bacteria. The findings are interesting and important.
Minor comments:
1) Fig. 5C is not indicated in the text.
2) Line 659: non-Ab should be non-nAb.
Author Response
We added Fig.5C to the third line on page 5 and changed non-Ab to non-nAb on line 658. We thank the referee for catching these errors.
Reviewer 2 Report
The study of Weiming Ouyang and colleagues compares the ability of Middle East Respiratory Syndrome (MERS), Severe Acute Respiratory Syndrome Coronavirus 1 (SARS CoV-1) and SARS CoV-2 spike proteins to induce cytokine expression in human peripheral blood mononuclear cells (PBMC). Like SARS CoV-1, SARS CoV-2 enters host cells via its spike protein, which attaches to angiotensin-converting enzyme 2 (ACE2) receptor. Authors report that some commercial lots of SARS CoV-2 spike protein-containing fusion proteins, but not lots of SARS CoV-1 or MERS spike protein-containing fusion proteins, stimulate human PBMC to produce proinflammatory cytokines. Induction of cytokine production by SARS CoV-2 spike fusion proteins consistently exhibits vendor and batch variability, and cytokine inductions are not blocked by soluble ACE2 or neutralizing anti-spike protein antibody. Moreover, Authors demonstrate that commercial spike protein reagents contained varying levels of endotoxin, which correlated directly with their abilities to induce cytokine production. The lipopolysaccharide (LPS) inhibitor, polymyxin B, blocked this cytokine induction activity. In addition, they demonstrate that SARS CoV-2 spike protein efficiently captures soluble LPS, rendering it capable of potent pro-inflammatory cytokine induction.
Authors conclude that these findings not only highlight the need to monitor LPS levels during in vitro and in vivo studies involving recombinant SARS CoV-2 spike protein, but indicate the possibility that interactions of SARS CoV-2 spike protein with LPS from commensal bacteria in virally infected mucosal tissues could promote pathogenic inflammatory cytokine production.
The study is interesting, but there are some issues to be clarified:
- One of the major issue is how Authors present the concentrations of the cytokines. I do not agree on the presentation as stimulation index, usually used to evaluate proliferation. The values of the cytokines must be expressed as pg or ng/ml.
- Fig.1 It is not clear. In the legend to fig. 1 Authors say that “Rested PBMC from two healthy donors (HD1 and HD2) were cultured ………………. Data shown are representative of the flow cytometric results from two independent experiments with total three healthy donors. Where are described results from the third healthy donor? Moreover in the results section, lines 123-124 “ To investigate the capability of SARS CoV-2 spike protein to directly induce a cytokine response, we cultured human PBMC from two healthy donors with a commercial SARS CoV-2 S1-Fc fusion protein ……”. Please explain these discrepancies
- Fig.2 line 637-639: The concentrations of the cytokines were calculated based on the standard curves, and the induction of cytokines were presented as stimulation indices. I do not understand why the values are not calculated as pg or ng/ml. It is important to know the amount of the cytokines expressed as pg or ng/ml. Stimulation index is usually used to evaluate proliferation. Please explain.
- References section: Ref. 8, 48 page number is missing; Ref. 21, 23, 26, 35, number of vol. and pages are missing
Author Response
1. One of the major issue is how Authors present the concentrations of the cytokines. I do not agree on the presentation as stimulation index, usually used to evaluate proliferation. The values of the cytokines must be expressed as pg or ng/ml.
Response: We agree with the referee that the presentation of cytokine inductions as stimulation indices has limitations in that it reflects only relative differences and not absolute concentrations. These values were calculated by dividing the cytokine amount of the treated group by that of the untreated group and used as the label for the Y axis. However, due to variable baseline cytokine levels observed in human PBMC from healthy donors, we still believe that there are advantages in demonstrating the data in this manner. However, to address the referee’s concern, we now also provide the concentrations of cytokines as supplementary tables (Table S1 for Fig. 2A and Fig. 3, Table S2 for Fig. 2B, Table S3 for Fig. 5E and Table S4 for Fig. 6B). In addition, cytokine inductions were recalculated using cytokine concentrations for Fig. 5E and Fig. 6B, as previous calculations were inadvertently made using MFI and not cytokine concentrations. We thank the referee for requesting this additional data to support our conclusions and provide supplemental information for the readers.
2. It is not clear. In the legend to fig. 1 Authors say that “Rested PBMC from two healthy donors (HD1 and HD2) were cultured ………………. Data shown are representative of the flow cytometric results from two independent experiments with total three healthy donors. Where are described results from the third healthy donor? Moreover in the results section, lines 123-124 “ To investigate the capability of SARS CoV-2 spike protein to directly induce a cytokine response, we cultured human PBMC from two healthy donors with a commercial SARS CoV-2 S1-Fc fusion protein ……”. Please explain these discrepancies
Response: We thank the referee for having identified these discrepancies. We used the S1-Fc lot 24056-2002-02 from Vendor #1 and PBMC from HD1 and HD2 for initial experiments, which were performed two or three times. An experiment using PBMC from the third donor was performed once with treatment using the same lot of S1-Fc protein (we had a limited supply of this reagent). To make it simple and consistent, we have removed the data from the third donor and change the figure legend to “data shown are representative of the flow cytometric results from at least two separate experiments performed on samples from healthy donors, HD1 and HD2”.
3. Fig.2 line 637-639: The concentrations of the cytokines were calculated based on the standard curves, and the induction of cytokines were presented as stimulation indices. I do not understand why the values are not calculated as pg or ng/ml. It is important to know the amount of the cytokines expressed as pg or ng/ml. Stimulation index is usually used to evaluate proliferation. Please explain.
Response: Please see our response to the first comment. Stimulation indices are replaced by cytokine inductions (fold), which is used for the label of Y axis. In addition, the absolute cytokine concentrations are presented in supplementary tables as requested.
4. References section: Ref. 8, 48 page number is missing; Ref. 21, 23, 26, 35, number of vol. and pages are missing.
Response: We thank the referee for careful checking of the references. We used Endnote to insert the references. For the identified references, there are no page numbers or no vol. and page numbers. We will discuss with the editor how to fix this issue.
Reviewer 3 Report
In this study, the authors studied the capabilities of SARS CoV-2 spike proteins from multiple commercial sources to induce cytokine productions of human peripheral blood mononuclear cells (PBMC). The conclusion is that only specific commercial lots of SARS CoV-2 induced cytokine production. However, the cytokine production was not mediated by SARS CoV-2 protein, instead, it was induced by LPS binds to SARS CoV-2 protein, so endotoxin level (LPS) of the commercial SARS CoV-2 proteins need to be considered when perform the in vitro assays.
Although the authors did plenty of in vitro assays with SARS CoV-2 proteins from different vendors and batches, the major concern of this study is lacking of novelty, as it had been published that SARS CoV-2 protein can binds to LPS and induce cytokines (J Mol Cell Biol 12, 916-932). Since the interaction between the LPS and SARS CoV-2 protein is already been studied, the present study is more like a technical report for testing the induction of inflammatory cytokines by SARS CoV-2 spike proteins from multiple commercial sources.
Author Response
Response: We thank the referee for having carefully reviewed our manuscript. We started this project well before publication of the paper reporting LPS binding to SARS CoV-2 spike protein (J Mol Cell Biol 12, 916-932), and we reached our conclusions via an entirely different approach. Although LPS binding with SARS CoV-2 spike protein was reported by this group (whom we cite), it is nevertheless surprising that high levels of LPS exist in spike protein reagents prepared using mammalian cell systems, which are being widely selected for studies in the SARS CoV-2 field. We have amended the text in the conclusion to include the sentence, “Unexpectedly, this high avidity binding has an adverse impact on conclusions generated from experiments involving some commercial SARS-CoV-2 spike protein reagents, even those manufactured in mammalian cell culture systems.”
Unfortunately, the impact of co-purifying LPS on the potential mechanisms of action of SARS CoV-2 spike protein is not fully appreciated. Even recently, papers and posters are reporting that inflammatory cytokine responses and lung tissue injury can be directly induced by SARS CoV-2 spike protein, which we believe, instead, are likely caused by the copurifying LPS in the reagents used in those studies. The misleading conclusions from these reports raise unnecessary safety concerns regarding SARS CoV-2 vaccination and may promote vaccine hesitancy. Therefore, we believe that this manuscript will provide novel and timely clarification pertinent to the understanding of SARS CoV-2 and its treatment. Especially in the setting of the current pandemic, the scientific community will benefit from the rapid communication not only of confounding factors underlying experiments in the published literature, but a likely pathogenic mechanism of SARS-CoV-2 as well.
Round 2
Reviewer 2 Report
Authors have addressed my concerns. Please correct typos errors in line 683
Reviewer 3 Report
No further comments.